# Utility metric for unsupervised feature selection

Amalia Villa[1,2], Abhijith Mundanad Narayanan[1,2],
Sabine Van Huffel[1,2], Alexander Bertrand[1,2] and Carolina Varon[1,3,4]

[1] STADIUS Center for Dynamical Systems, Signal Processing and Data Analytics, Department of Electrical Engineering (ESAT), KU Leuven, Leuven, Belgium
[2] Leuven.AI, KU Leuven Institute for AI, Leuven, Belgium
[3] Circuits and Systems (CAS) Group, Delft University of Technology, Delft, The Netherlands
[4] e-Media Research Lab, Campus GroepT, KU Leuven, Leuven, Belgium



Corresponding author
Amalia Villa,
amalia.villagomez@kuleuven.be

## ABSTRACT

Feature selection techniques are very useful approaches for dimensionality reduction in data analysis. They provide interpretable results by reducing the dimensions of the data to a subset of the original set of features. When the data lack annotations, unsupervised feature selectors are required for their analysis. Several algorithms for this aim exist in the literature, but despite their large applicability, they can be very inaccessible or cumbersome to use, mainly due to the need for tuning non-intuitive parameters and the high computational demands. In this work, a publicly available ready-to-use unsupervised feature selector is proposed, with comparable results to the state-of-the-art at a much lower computational cost. The suggested approach belongs to the methods known as spectral feature selectors. These methods generally consist of two stages: manifold learning and subset selection. In the first stage, the underlying structures in the high-dimensional data are extracted, while in the second stage a subset of the features is selected to replicate these structures. This paper suggests two contributions to this field, related to each of the stages involved. In the manifold learning stage, the effect of non-linearities in the data is explored, making use of a radial basis function (RBF) kernel, for which an alternative solution for the estimation of the kernel parameter is presented for cases with high-dimensional data. Additionally, the use of a backwards greedy approach based on the least-squares utility metric for the subset selection stage is proposed. The combination of these new ingredients results in the utility metric for unsupervised feature selection U2FS algorithm. The proposed U2FS algorithm succeeds in selecting the correct features in a simulation environment. In addition, the performance of the method on benchmark datasets is comparable to the state-of-the-art, while requiring less computational time. Moreover, unlike the state-of-the-art, U2FS does not require any tuning of parameters.

# INTRODUCTION

Many applications of data science require the study of highly multi-dimensional data. A high number of dimensions implies a high computational cost as well as a large amount

of memory required. Furthermore, this often leads to problems related to the curse of dimensionality (*Verleysen & François, 2005*) and thus, to irrelevant and redundant data for machine learning algorithms (*Maindonald, 2007*). Therefore, it is crucial to perform dimensionality reduction before analyzing the data.

There are two types of dimensionality reduction techniques. So-called feature selection techniques directly select a subset of the original features. On the other hand, transformation techniques compute a new (smaller) set of features, each of which are derived from all features of the original set. Some examples of these are Principal Component Analysis (PCA) (*Wold, Esbensen & Geladi, 1987*), Independent Component Analysis (ICA) (*Jiang et al., 2006*) or the Extended Sammon Projection (ESP) (*Ahmad et al., 2019*). While these methods lead to a reduction in the number of dimensions, results are less interpretable, since their direct relationship with the original set of features is lost.

In this work, the focus is on unsupervised feature selectors. Since these methods do not rely on the availability of labels or annotations in the data, the information comes from the learning of the underlying structure of the data. Despite this challenge, the generalization capabilities of these methods are typically better than for supervised or semi-supervised methods (*Guyon & Elisseeff, 2003*). Within unsupervised feature selectors, sparse learning based methods have gained attention in the last 20 years (*Li et al., 2017*). These methods rely on graph theory and manifold learning to learn the underlying structures of the data (*Lunga et al., 2013*), and they apply sparsity inducing techniques to perform subset selection. However, to the best of our knowledge, none explores specifically the behavior of these methods with data presenting non-linear relationships between the features (i.e., dimensions). While the graph definition step can make use of kernels to tackle non-linearities, these can be heavily affected by the curse of dimensionality, since they are often based on a distance metric (*Aggarwal, Hinneburg & Keim, 2001*).

After the manifold learning stage, sparse regression is applied to score the relevance of the features in the structures present in the graph. These formulations make use of sparsity-inducing regularization techniques to provide the final subset of features selected, and thus, they are highly computationally expensive. These methods are often referred to as structured sparsity-inducing feature selectors (SSFS), or sparse learning based methods (*Gui et al., 2016*; *Li et al., 2017*).

Despite the large amount of unsupervised SSFS algorithms described in the literature, these methods are cumbersome to use for a novice user. This is not only due to the codes not being publicly available, but also due to the algorithms requiring regularization parameters which are difficult to tune, in particular in unsupervised settings.

In this work, an efficient unsupervised feature selector based on the utility metric (U2FS) is proposed. U2FS is a ready-to-use, publicly available unsupervised sparsity-inducing feature selector designed to be robust for data containing non-linearities.

The code is available here: https://github.com/avillago/u2fs, where all functions and example codes are published. The main contributions of this work are:

- The definition of a new method to automatically approximate the radial-basis function (RBF) kernel parameter without the need for a user-defined tuning parameter. This method is used to tackle the curse of dimensionality when embedding the data taking non-linearities into account.

- The suggestion of a backwards greedy approach for the stage of subset selection, based on the utility metric for the least-squares problem. The utility metric was proposed in the framework of supervised learning (*Bertrand, 2018*), and has been used for channel selection in applications such as electroencephalography (EEG) (*Narayanan & Bertrand, 2020*), sensor networks (*Szurley et al., 2014*), and microphone arrays (*Szurley, Bertrand & Moonen, 2012*). Nevertheless, this is the first work in which this type of approach is proposed for the sparsity-inducing stage of feature selection.

- Propose a non-parametric and efficient unsupervised SSFS algorithm. This work analyzes the proposed method U2FS in terms of its complexity, and of its performance on simulated and benchmark data. The goal is to reduce the computational cost while maintaining a comparable performance with respect to the state-of-the-art. In order to prove this, U2FS is compared to three related state-of-the-art algorithms in terms of accuracy of the features selected, and computational complexity of the algorithm.

The rest of the paper is structured as follows. In Related Work, previous algorithms on SSFS are summarized. In Methods, the proposed U2FS method is described: first the manifold learning stage, together with the algorithm proposed for the selection of the kernel parameter; and further on, the utility metric is discussed and adapted to feature selection. The experiments performed in simulations and benchmark databases, as well as the results obtained are described in the Results and Discussion sections. Finally, the last section provides some conclusions.

## RELATED WORK

Sparsity-inducing feature selection methods have become widely used in unsupervised learning applications for high-dimensional data. This is due to two reasons. On the one hand, the use of manifold learning guarantees the preservation of local structures present in the high-dimensional data. Additionally, its combination with feature selection techniques not only reduces the dimensionality of the data, but also guarantees interpretability.

Sparsity-inducing feature selectors learn the structures present in the data via connectivity graphs obtained in the high-dimensional space (*Yan et al., 2006*). The combination of manifold learning and regularization techniques to impose sparsity, allows to select a subset of features from the original dataset that are able to describe these structures in a smaller dimensional space.

These algorithms make use of sparsity-inducing regularization approaches to stress those features that are more relevant for data separation. The sparsity of these approaches is controlled by different statistical norms ($l_{r,p}$-norms), which contribute to the generalization capability of the methods, adapting them to binary or multi-class problems (*Gui et al., 2016*). One drawback of these sparse regression techniques is that generally, they rely on optimization methods, which are computationally expensive.

The Laplacian Score (*He, Cai & Niyogi, 2006*) was the first method to perform spectral feature selection in an unsupervised way. Based on the Laplacian obtained from the spectral embedding of the data, it obtains a score based on locality preservation. SPEC (*Zhao & Liu, 2007*) is a framework that contains this previous approach, but it additionally allows for both supervised or unsupervised learning, including other similarity metrics, as well as other ranking functions. These approaches evaluate each feature independently, without considering feature interactions. These interactions are, however, taken into account in Multi-Cluster Feature Selection (MCFS) (*Cai, Zhang & He, 2010*), where a multi-cluster approach is defined based on the eigendecomposition of a similarity matrix. The subset selection is performed applying an $l_1$-norm regularizer to approximate the eigenvectors obtained from the spectral embedding of the data inducing sparsity. In UDFS (*Yang et al., 2011*) the $l_1$-norm regularizer is substituted by a $l_{2,1}$-norm to apply sample and feature-wise constraints, and a discriminative analysis is added in the graph description. In NDFS (*Li et al., 2012*), the use of the $l_{2,1}$-norm is preserved, but a non-negative constraint is added to the spectral clustering stage. Additionally, this algorithm performs feature selection and spectral clustering simultaneously.

The aforementioned algorithms perform manifold learning and subset selection in a sequential way. However, other methods tackle these simultaneously, in order to adaptively change the similarity metric or the selection criteria regarding the error obtained between the original data and the new representation. Examples of these algorithms are JELSR (*Hou et al., 2013*), SOGFS (*Nie, Zhu & Li, 2019*), (R)JGSC (*Zhu et al., 2016*) and DSRMR (*Tang et al., 2018*), and all make use of an $l_{2,1}$-norm. Most recently, the SAMM-FS algorithm was proposed (*Zhang et al., 2019*), where a combination of similarity measures is used to build the similarity graph, and the $l_{2,0}$-norm is used for regression. This group of algorithms are currently the ones achieving the best results, at the cost of using complex optimization techniques to adaptively tune both stages of the feature selection process. While this can lead to good results, it comes with a high computation cost, which might hamper the tuning process, or might simply not be worthy for some applications. SAMM-FS and SOGFS are the ones that more specifically suggest new approaches to perform the embedding stage, by optimally creating the graph (*Nie, Zhu & Li, 2019*) or deriving it from a combination of different similarity metrics (*Zhang et al., 2019*). Again, both approaches require computationally expensive optimization techniques to select a subset of features.

In summary, even if SSFS methods are getting more sophisticated and accurate, this results in algorithms becoming more complex in terms of computational time, and in the ease of use. The use of advanced numerical optimization techniques to improve results makes algorithms more complex, and requires regularization parameters which are not

easy to tune. In this work, the combination of a new approach to estimate the graph connectivity based on the RBF kernel, together with the use of the utility metric for subset selection, results in an efficient SSFS algorithm, which is easy to use and with lower complexity than the state-of-the-art. This efficient implementation is competitive with state-of-the-art methods in terms of performance, while using a simpler strategy, which is faster to compute and easier to use.

## METHODS

This section describes the proposed U2FS algorithm, which focuses on selecting the relevant features in an unsupervised way, at a relatively small computational cost. The method is divided in three parts. Firstly, the suggested manifold learning approach is explained, where an embedding based on binary weighting and the RBF kernel are used. Then a method to select the kernel parameter of the RBF kernel is proposed, specially designed for high-dimensional data. Once the manifold learning stage is explained, the Utility metric is proposed as a new approach for subset selection.

### Manifold learning considering non-linearities

Given is a data matrix $\mathbf{X} \in \mathbb{R}^{N \times d}$, with $\mathbf{X} = [\mathbf{x}_1; \mathbf{x}_2; \ldots; \mathbf{x}_N]$, $\mathbf{x}_i = [x_i^{(1)}, x_i^{(2)}, \ldots, x_i^{(d)}]$, $i = 1, \ldots, N$, $N$ the number of data points, and $d$ the number of features (i.e., dimensions) in the data. The aim is to learn the structure hidden in the $d$-dimensional data and approximate it with only a subset of the original features. In this paper, this structure will be identified by means of clustering, where the dataset is assumed to be characterized by $c$ clusters.

In spectral clustering, the clustering structure of this data can be obtained by studying the eigenvectors derived from a Laplacian built from the original data (*Von Luxburg (2007)*, *Biggs, Biggs & Norman (1993)*). The data is represented using a graph $G = (\mathcal{V}, \mathcal{E})$. $\mathcal{V}$ is the set of vertices $\mathbf{v}_i$, $i = 1, \ldots, N$ where $\mathbf{v}_i = \mathbf{x}_i$. $\mathcal{E} = \{e_{ij}\}$ with $i = 1, \ldots, N$ $j = 1, \ldots, N$ is the set of edges between the vertices where $\{e_{ij}\}$ denotes the edge between vertices $v_i$ and $v_j$. The weight of these edges is determined by the entries $w_{ij} \geq 0$ of a similarity matrix $\mathbf{W}$. We define the graph as undirected. Therefore, the similarity matrix $\mathbf{W}$, is symmetric (since $w_{ij} = w_{ji}$, with the diagonal set to $w_{ii} = 0$).

Typically, $\mathbf{W}$ is computed after coding the pairwise distances between all $N$ data points. There are several ways of doing this, such as calculating the $k$-nearest neighbours (KNN) for each point, or choosing the $\varepsilon$-neighbors below a certain distance (*Belkin & Niyogi, 2002*).

In this paper, two similarity matrices are adopted inspired by the work in (*Cai, Zhang & He, 2010*), namely a binary one and one based on an RBF kernel. The binary weighting is based on KNN, being $w_{ij} = 1$ if and only if vertex $i$ is within the $K$ closest points to vertex $j$. Being a non-parametric approach, the binary embedding allows to simply characterize the connectivity of the data.

Additionally, the use of the RBF kernel is considered, which is well suited for non-linearities and allows to characterize complex and sparse structures (*Von Luxburg, 2007*). The RBF kernel is defined as $K(\mathbf{x}_i, \mathbf{x}_j) = exp(-||\mathbf{x}_i - \mathbf{x}_j||^2/2\sigma^2)$. The selection of the

kernel parameter $\sigma$ is a long-standing challenge in machine learning. For instance, in *Cai, Zhang & He (2010)*, $\sigma^2$ is defined as the mean of all the distances between the data points. Alternatively, a rule of thumb, uses the sum of the standard deviations of the data along each dimension (*Varon, Alzate & Suykens, 2015*). However, the estimation of this parameter is highly influenced by the amount of features or dimensions in the data, making it less robust to noise and irrelevant features. In the next section, a new and better informed method to approximate the kernel parameter is proposed.

The graph $G$, defined by the similarity matrix $\mathbf{W}$, can be partitioned into multiple disjoint sets. Given the focus on multi-cluster data of our approach, the $k$-Way Normalized Cut (*NCut*) Relaxation is used, as proposed in *Ng, Jordan & Weiss (2002)*. In order to obtain this partition, the degree matrix $\mathbf{D}$ of $\mathbf{W}$ must be calculated. $\mathbf{D}$ is a diagonal matrix for which each element on the diagonal is calculated as $D_{ii} = \sum_j W_{i,j}$. The normalized Laplacian $\mathbf{L}$ is then obtained as $\mathbf{L} = \mathbf{D}^{-1/2}\mathbf{W}\mathbf{D}^{-1/2}$, as suggested in *Von Luxburg (2007)*. The vectors $\mathbf{y}$ embedding the data in $\mathbf{L}$ can be extracted from the eigenvalue problem (*Chung & Graham, 1997*):

$$\mathbf{L}\mathbf{y} = \lambda\mathbf{y} \tag{1}$$

Given the use of a normalized Laplacian for the data embedding, the vectors $\mathbf{y}$ must be adjusted using the degree matrix $\mathbf{D}$:

$$\alpha = \mathbf{D}^{1/2}\mathbf{y}, \tag{2}$$

which means that $\alpha$ is the solution of the generalized eigenvalue problem of the pair $\mathbf{W}$ and $\mathbf{D}$. These eigenvectors $\alpha$ are a new representation of the data, that gathers the most relevant information about the structures appearing in the high-dimensional space. The $c$ eigenvectors, corresponding to the $c$ highest eigenvalues (after excluding the largest one), can be used to characterize the data in a lower dimensional space (*Ng, Jordan & Weiss, 2002*). Thus, the matrix $\mathbf{E} = [\alpha_1, \alpha_2, \ldots, \alpha_c]$ containing column-wise the $c$ selected eigenvectors, will be the low-dimensional representation of the data to be mimicked using a subset of the original features, as suggested in *Cai, Zhang & He (2010)*.

## Kernel parameter approximation for high-dimensional data

One of the most used similarity functions is the RBF kernel, which allows to explore non-linearities in the data. Nevertheless, the kernel parameter $\sigma^2$ must be selected correctly, to avoid overfitting or the allocation of all data points to the same cluster. This work proposes a new approach to approximate this kernel parameter, which will be denoted by $\hat{\sigma}^2$ when derived from our method. This method takes into account the curse of dimensionality and the potential irrelevant features or dimensions in the data.

As a rule of thumb, $\sigma^2$ is approximated as the sum of the standard deviation of the data along each dimension (*Varon, Alzate & Suykens, 2015*). This approximation grows with the number of features (i.e., dimensions) of the data, and thus, it is not able to capture its underlying structures in high-dimensional spaces. Nevertheless, this $\sigma^2$ is commonly used as an initialization value, around which a search is performed, considering some objective function (*Alzate & Suykens, 2008*; *Varon, Alzate & Suykens, 2015*).

The MCFS algorithm skips the search around an initialization of the $\sigma^2$ value by substituting the sum of the standard deviations by the mean of these (*Cai, Zhang & He, 2010*). By doing so, the value of $\sigma^2$ does not overly grow. This estimation of $\sigma^2$ suggested in *Cai, Zhang & He (2010)* will be referred to as $\sigma_0^2$. A drawback of this approximation in high-dimensional spaces is that it treats all dimensions as equally relevant for the final estimation of $\sigma^2{}_0$, regardless of the amount of information that they actually contain.

The aim of the proposed approach is to provide a functional value of $\sigma^2$ that does not require any additional search, while being robust to high-dimensional data. Therefore, this work proposes an approximation technique based on two factors: the distances between the points, and the number of features or dimensions in the data.

The most commonly used distance metric is the euclidean distance. However, it is very sensitive to high-dimensional data, deriving unsubstantial distances when a high number of features is involved in the calculation (*Aggarwal, Hinneburg & Keim, 2001*). In this work, the use of the Manhattan or taxicab distance (*Reynolds, 1980*) is proposed, given its robustness when applied to high-dimensional data (*Aggarwal, Hinneburg & Keim, 2001*). For each feature $l$, the Manhattan distance $\delta_l$ is calculated as:

$$\delta_l = \frac{1}{N} \sum_{i,j=1}^{N} |x_{il} - x_{jl}| \tag{3}$$

Additionally, in order to reduce the impact of irrelevant or redundant features, a system of weights is added to the approximation of $\hat{\sigma}^2$. The goal is to only take into account the distances associated to features that contain relevant information about the structure of the data. To calculate these weights, the probability density function (PDF) of each feature is compared with a Gaussian distribution. Higher weights are assigned to the features with less Gaussian behavior, i.e., those the PDF of which differs the most from a Gaussian distribution. By doing so, these will influence more the final $\hat{\sigma}^2$ value, since they allow a better separation of the structures present in the data.

Figure 1 shows a graphical representation of this estimation. The dataset in the example has 3 dimensions or features: $f_1$, $f_2$ and $f_3$. $f_1$ and $f_2$ contain the main clustering information, as it can be observed in Fig. 1A, while $f_3$ is a noisy version of $f_1$, derived as $f_3 = f_1 + 1.5n$, where $n$ is drawn from a normal distribution $\mathcal{N}(0, 1)$. Figs. 1B, 1C and 1D show in a continuous black line the PDFs derived from the data, and in a grey dash line their fitted Gaussian, in dimensions $f_1$, $f_2$ and $f_3$ respectively. This fitted Gaussian was derived using the Curve Fitting toolbox of Matlab™. As it can be observed, the matching of a Gaussian with an irrelevant feature is almost perfect, while those features that contain more information, like $f_1$ and $f_2$, deviate much more from a normal distribution.

Making use of these differences, an error, denoted $\phi_l$, for each feature $l$, where $l = 1, \dots, d$, is calculated as:

$$\phi_l = \frac{1}{H} \sum_{i=1}^{H} (p_i - g_i)^2, \tag{4}$$

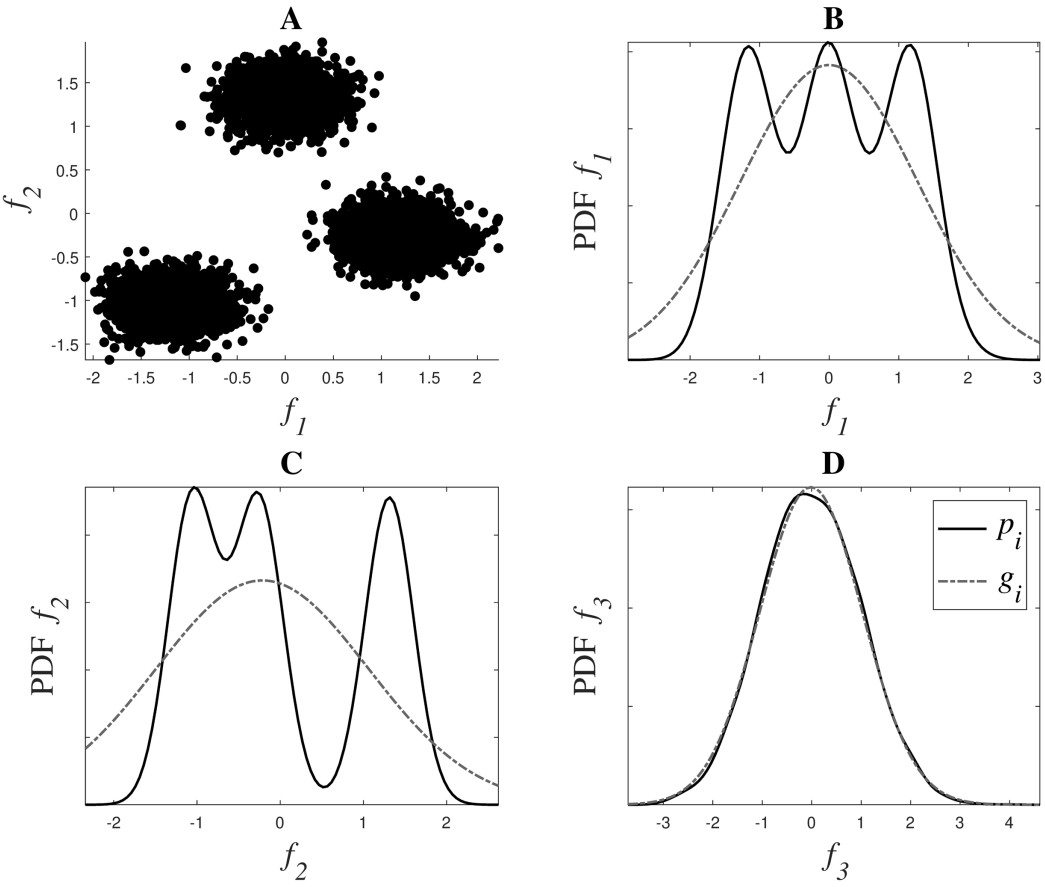

**Figure 1** **Weight system for relevance estimation. In (A), $f_1$ and $f_2$ can be seen. (B, C and D) show in black the PDFs $p_i$ of $f_1$, $f_2$ and $f_3$ respectively, and in grey dotted line their fitted Gaussian $g_i$.**

where $H$ is the number of bins in which the range of the data is divided to estimate the PDF ($p$), and $g$ is the fitted Gaussian. The number of bins in this work is set to 100 for standardization purposes. Equation (4) corresponds to the mean-squared error (MSE) between the PDF of the data over feature $l$ and its fitted Gaussian. From these $\phi_l$, the final weights $b_l$ are calculated as:

$$b_l = \frac{\phi_l}{\sum_{l=1}^{d} \phi_l} \tag{5}$$

Therefore, combining (3) and (5), the proposed approximation, denoted $\hat{\sigma}^2$, is derived as:

$$\hat{\sigma}^2 = \sum_{l=1}^{d} b_l \delta_l, \tag{6}$$

which gathers the distances present in the most relevant features, giving less importance to the dimensions that do not contribute to describe the structure of the data. The complete algorithm to calculate $\hat{\sigma}^2$ is described in Algorithm 1.

---

**Algorithm 1** Kernel parameter approximation for high-dimensional data.

**Input:** Data $\mathbf{X} \in \mathbb{R}^{N \times d}$.

**Output:** Sigma parameter $\hat{\sigma}^2$

1: Calculate the Manhattan distances between the datapoints using Equation (3): vector of distances per feature $\delta_l$.

2: Obtain the weights for each of the features using Equations (4) and (5): weights $b_l$.

3: Calculate $\hat{\sigma}^2$ using Equation (6).

---

## Utility metric for feature subset selection

In the manifold learning stage, a new representation $\mathbf{E}$ of the data based on the eigenvectors was built, which described the main structures present in the original high-dimensional data. The goal is to select a subset of the features which best approximates the data in this new representation. In the literature, this feature selection problem is formulated using a graph-based loss function and a sparse regularizer of the coefficients is used to select a subset of features, as explained in *Zhu et al. (2016)*. The main idea of these approaches is to regress the data to its low dimensional embedding along with some sparse regularization. The use of such regularization techniques reduces overfitting and achieves dimensionality reduction. This regression is generally formulated as a least squares (LS) problem, and in many of these cases, the metric that is used for feature selection is the magnitude of their corresponding weights in the least squares solution (*Cai, Zhang & He, 2010*; *Gui et al., 2016*). However, the optimized weights do not necessarily reflect the importance of the corresponding feature as it is scaling dependent and it does not properly take interactions across features into account (*Bertrand, 2018*). Instead, the importance of a feature can be quantified using the increase in least-squared error (LSE) if that feature was to be removed and the weights were re-optimized. This increase in LSE, called the 'utility' of the feature can be efficiently computed (*Bertrand, 2018*) and can be used as an informative metric for a greedy backwards feature selection procedure (*Bertrand, 2018*; *Narayanan & Bertrand, 2020*; *Szurley et al., 2014*), as an alternative for (group-) LASSO based techniques. Under some technical conditions, a greedy selection based on this utility metric can even be shown to lead to the optimal subset (*Couvreur & Bresler, 2000*).

After representing the dataset using the matrix $\mathbf{E} \in \mathbb{R}^{N \times c}$ containing the $c$ eigenvectors, the following LS optimization problem finds the weights $\mathbf{p}$ that best approximate the data $\mathbf{X}$ in the $c$-dimensional representation in $\mathbf{E}$:

$$J = \min_{\mathbf{P}} \frac{1}{N} ||\mathbf{Xp} - \mathbf{E}||_F^2 \tag{7}$$

where $J$ is the cost or the LSE and $||.||_F$ denotes the Frobenius norm.

If $\mathbf{X}$ is a full rank matrix and if $N > d$, the LS solution $\hat{\mathbf{p}}$ of (7) is

$$\hat{\mathbf{p}} = \mathbf{R}_{\mathbf{XX}}^{-1}\mathbf{R}_{\mathbf{XE}}, \tag{8}$$

with $\mathbf{R}_{\mathbf{XX}} = \frac{1}{N}\mathbf{X}^T\mathbf{X}$ and $\mathbf{R}_{\mathbf{XE}} = \frac{1}{N}\mathbf{X}^T\mathbf{E}$.

The goal of this feature selection method is to select the subset of $s(<d)$ features that best represents $\mathbf{E}$. This feature selection problem can be reduced to the selection of the best $s(<d)$ columns of $\mathbf{X}$ which minimize (7). However, this is inherently a combinatorial problem and is computationally unfeasible to solve. Nevertheless, several greedy and approximative methods have been proposed (*Gui et al., 2016*; *Nie, Zhu & Li, 2019*; *Narayanan & Bertrand, 2020*). In the current work, the use of the utility metric for subset selection is proposed to select these best $s$ columns.

The utility of a feature $l$ of $\mathbf{X}$, in an LS problem like (7), is defined as the increase in the LSE $J$ when the column corresponding to the $l$-th feature in $\mathbf{X}$ is removed from the problem and the new optimal weight matrix, $\hat{\mathbf{p}}_{-l}$, is re-computed similar to (8). Consider the new LSE after the removal of feature $l$ and the re-computation of the weight matrix $\hat{\mathbf{p}}_{-l}$ to be $J_{-l}$, defined as:

$$J_{-l} = \frac{1}{N}||\mathbf{X}_{-l}\hat{\mathbf{p}}_{-l} - \mathbf{E}||_F^2 \tag{9}$$

where $\mathbf{X}_{-l}$ denotes the matrix $\mathbf{X}$ with the column corresponding to $l$-th feature removed. Then according to the definition, the utility of feature $l$, $U_l$ is:

$$U_l = J_{-l} - J \tag{10}$$

A straightfoward computation of $U_l$ would be computationally heavy due to the fact that the computation of $\hat{\mathbf{p}}_l$ requires a matrix inversion of $\mathbf{X}_{-l}\mathbf{X}_{-l}^T$, which has to be repeated for each feature $l$.

However, it can be shown that the utility of the $l$-th feature of $\mathbf{X}$ in (10) can be computed efficiently without the explicit recomputation of $\hat{\mathbf{p}}_{-l}$ by using the following expression (*Bertrand, 2018*):

$$U_l = \frac{1}{q_l}||\bar{\mathbf{p}}_l||_2, \tag{11}$$

where $q_l$ is the $l$-th diagonal element of $\mathbf{R}_{XX}^{-1}$ and $p_l$ is the $l$-th row in $\hat{\mathbf{p}}$, corresponding to the $l$-th feature. The mathematical proof of (11) can be found in *Bertrand (2018)*. Note that $\mathbf{R}_{XX}^{-1}$ is already known from the computation of $\hat{\mathbf{p}}$ such that no additional matrix inversion is required.

However, since the data matrix $\mathbf{X}$ can contain redundant features or features that are linear combinations of each other in its columns, it cannot be guaranteed that the matrix $\mathbf{X}$ in (7) is full-rank. In this case, the removal of a redundant column from $\mathbf{X}$ will not lead to an increase in the LS cost of (7). Moreover, $\mathbf{R}_{XX}^{-1}$, used to find the solution of (7) in (8), will not exist in this case since the matrix $\mathbf{X}$ is rank deficient. A similar problem appears if $N < d$, which can happen in case of very high-dimensional data. To overcome this problem, the definition of utility generalized to a minimum $l_2$-norm selection (*Bertrand, 2018*) is used in this work. This approach eliminates the feature yielding the smallest increase in the $l_2$-norm of the weight matrix when the column corresponding to that feature were to be removed and the weight matrix would be re-optimized. Moreover, minimizing the $l_2$-norm of the weights further reduces the risk of overfitting.

This generalization is achieved by first adding an $l_2$-norm penalty $\beta$ to the cost function that is minimized in (7):

$$J = \min_{\mathbf{p}} \frac{1}{2} ||\mathbf{Xp} - \mathbf{E}||_F^2 + \beta ||\mathbf{p}||_2^2 \qquad (12)$$

where $0 < \beta$ $\mu$ with $\mu$ equal to the smallest non-zero eigenvalue of $\mathbf{R_{XX}}$ in order to ensure that the bias added due to the penalty term in (12) is negligible. The minimizer of (12) is:

$$\hat{\mathbf{p}} = \mathbf{R_{XX\beta}^{-1}} \mathbf{R_{XE}} = (\mathbf{R_{XX}} + \beta \mathbf{I})^{-1} \mathbf{R_{XE}} \qquad (13)$$

It is noted that (13) reduces to $\mathbf{R_{XX}^{\dagger}} \mathbf{R_{XE}}$ when $\beta \to 0$, where $\mathbf{R_{XX}^{\dagger}}$ denotes the Moore-Penrose pseudo-inverse. This solution corresponds to the minimum norm solution of (7) when $\mathbf{X}$ contains linearly dependent columns or rows. The utility $U_l$ of the $l$-th column in $\mathbf{X}$ based on (12) is (*Bertrand, 2018*):

$$\begin{aligned} U_l &= \left( ||\mathbf{X}_{-l}\hat{\mathbf{p}}_{-l} - \mathbf{E}||_2^2 - ||\mathbf{X}\hat{\mathbf{p}} - \mathbf{E}||_2^2 \right) \\ &\quad + \beta \left( ||\hat{\mathbf{p}}_{-l}||_2^2 - ||\hat{\mathbf{p}}||_2^2 \right) \\ &= (J_{-l} - J) + \beta \left( ||\hat{\mathbf{p}}_{-l}||_2^2 - ||\hat{\mathbf{p}}||_2^2 \right) \end{aligned} \qquad (14)$$

Note that if column $l$ in $\mathbf{X}$ is linearly independent from the other columns, (14) closely approximates to the original utility definition in (10) as the first term dominates over the second. However, if column $l$ is linearly dependent, the first term vanishes and the second term will dominate. In this case, the utility quantifies the increase in $l_2$-norm after removing the $l$-th feature.

To select the best $s$ features of $\mathbf{X}$, a greedy selection based on the iterative elimination of the features with the least utility is carried out. After the elimination of each feature, a re-estimation of the weights $\hat{\mathbf{p}}$ is carried out and the process of elimination is repeated, until $s$ features remain.

Note that the value of $\beta$ depends on the smallest non-zero eigenvalue of $\mathbf{R_{XX}}$. Since $\mathbf{R_{XX}}$ has to be recomputed every time when a feature is removed, also its eigenvalues change along the way. In practice, the value of $\beta$ is selected only once and fixed for the remainder of the algorithm, as smaller than the smallest non-zero eigenvalue of $\mathbf{R_{XX}}$ before any of the features are eliminated (*Narayanan & Bertrand, 2020*). This value of $\beta$ will be smaller than all the non-zero eigenvalues of any principal submatrix of $\mathbf{R_{XX}}$ using the Cauchy's interlace theorem (*Hwang, 2004*).

The summary of the utility subset selection is described in Algorithm 2. Algorithm 3 outlines the complete U2FS algorithm proposed in this paper.

As it has been stated before, one of the most remarkable aspects of the U2FS algorithm is the use of a greedy technique to solve the subset selection problem. The use of this type of method reduces the computational cost of the algorithm. This can be confirmed analyzing the computational complexity of U2FS, where the most demanding steps are the eigendecomposition of the Laplacian matrix (step 2 of Algortihm 4), which has a cost of $O(N^3)$ (*Tsironis et al., 2013*), and the subset selection stage in step 3 of Algorithm 4. Contrary to the state-of-the-art, the complexity of U2FS being a greedy method depends on the number of features to select. The most computationally expensive step of the subset

---

**Algorithm 2** Utility metric algorithm for subset selection.

**Input:** Data $\mathbf{X}$, Eigenvectors $\mathbf{E}$, Number of features $s$ to select

**Output:** $s$ features selected

1: Calculate $\mathbf{R_{XX}}$ and $\mathbf{R_{XE}}$ as described in Equation (8).

2: Calculate $\beta$ as the smallest non-zero eigenvalue of $\mathrm{R_{XX}}$

3: **while** Number of features remaining is $> s$ **do**

4: Compute $\mathrm{R}_{\mathbf{XX_\beta}}^{-1}$ and $\hat{\mathbf{p}}$ as described in (13).

5: Calculate the utility of the remaining features using (11)

6: Remove the feature $f_l$ with the lowest utility.

7: Update $\mathbf{R_{XX}}$ and $\mathbf{R_{XE}}$ by removing the rows and columns related to that feature $f_l$.

8: **end while**

---

**Algorithm 3** Unsupervised feature selector based on the utility metric (U2FS).

**Input:** Data $\mathbf{X}$, Number of clusters $c$, Number of features $s$ to select

**Output:** $s$ features selected

1: Construct the similarity graph $\mathbf{W}$ as described in Section selecting one of the weightings:

- Binary

- RBF kernel, using $\sigma_0^2$

- RBF kernel, using $\hat{\sigma}^2$ based on Algorithm 1

2: Calculate the normalized Laplacian $\mathbf{L}$ and the eigenvectors $\alpha$ derived from Equation (2). Keep the $c$ eigenvectors corresponding to the highest eigenvalues, excluding the first one.

3: Apply the backward greedy utility algorithm 2.

4: Return the s features remaining from the backward greedy utility approach.

---

selection in U2FS is the calculation of the matrix $\mathbf{R_{XX}^{-1}}$, which has a computational cost of $O(d^3)$. In addition, this matrix needs to be updated $d - s$ times. This update can be done efficiently using a recursive updating equation from *Bertrand (2018)* with a cost of $O(t^2)$, with $t$ the number of features remaining in the dataset, i.e., $t = d - s$. Since $t < d$, the cost for performing $d - s$ iterations will be $O((d - s)d^2)$, which depends on the number of features $s$ to be selected. Note that the cost of computing the least squares solution $\hat{\mathbf{p}}_{-l}$ for each $l$ in (14) is eliminated using the efficient Eq. (11), bringing down the cost for computing the utility from $O(t^4)$ to $O(t)$ in each iteration. This vanishes with respect to the $O(d^3)$ term (remember that $t < d$). Therefore, the total asymptotic complexity of U2FS is $O(N^3 + d^3)$.

## RESULTS

The aim of the following experiments is to evaluate the U2FS algorithm based on multiple criteria. With the focus on the new estimation of the embedding proposed, the proposed RBF kernel approach using the estimated $\hat{\sigma}^2$ is compared to the $\sigma_0^2$ parameter proposed in *Cai, Zhang & He (2010)*, and to the binary KNN graph commonly used in *Gui et al. (2016)*. On the other hand, the utility metric for subset selection is compared

---

**Table 1  Methods compared in the experiments.**

| | Similarity measure | Subset selection |
|---|---|---|
| $\text{KNN}_{Bin} + l_1 - \text{norm}$ | KNN + binary weighting | $l_1$-norm |
| $\text{RBF}_{\sigma_0^2} + l_1 - \text{norm}$ | RBF kernel, $\sigma_0^2$ | $l_1$-norm |
| $\text{KNN}_{Bin} + \text{Utility}$ | KNN + binary weighting | Utility metric |
| $\text{RBF}_{\sigma_0^2} + \text{Utility}$ | RBF kernel, $\sigma_0^2$ | Utility metric |
| $\text{RBF}_{\hat{\sigma}^2} + \text{Utility}$ | RBF kernel, $\hat{\sigma}^2$ | Utility metric |

to other sparsity-inducing techniques, based on $l_p - norm$ regularizations. In these experiments, this is evaluated using the $l_1 - norm$. The outline of the different combinations considered in this work summarized in Table 1. The last method, $\text{RBF}_{\hat{\sigma}^2} + \text{Utility}$, would be the one referred to as U2FS, combining the novelties suggested in this work.

These novelties are evaluated in two different scenarios, namely a simulation study, and in the application of the methods on benchmark datasets. In particular for the latter, the methods are not only evaluated in terms of accuracy, but also regarding computational cost. Additionally, U2FS is compared with 3 representative state-of-the-art algorithms from the general family of unsupervised sparsity-inducing feature selection algorithms:

- **MCFS** (*Cai, Zhang & He, 2010*) (http://www.cad.zju.edu.cn/home/dengcai/Data/MCFS.html). This algorithm served as inspiration to create U2FS, and therefore, it is added to the set of comparison algorithms as baseline reference. MCFS performs spectral embedding and $l_1$-norm regularization sequentially, and which served as inspiration to create U2FS.

- **NDFS** (*Li et al., 2012*) (http://www.cs.cmu.edu/yiyang/Publications.html), which performs nonnegative spectral analysis with $l_{2,1}$-norm regularization. This algorithm is added to the experiments since it is an improvement of MCFS, while being the first algorithm simultaneously adapting both stages of manifold learning and subset selection. Therefore, NDFS represents the transition to these adaptive optimization-based feature selection algorithms.

- **RJGSC** (*Zhu et al., 2016*) optimally derives the embedding of the data by adapting the results with $l_{2,1}$-norm regularization. This algorithm is taken as a reference for the large class of adaptive sparsity-inducing feature selection algorithms, which are much more complex than U2FS, since they apply optimization to recursively adapt the embedding and feature selection stages of the methods. RJGSC was already compared to several feature selectors in *Zhu et al. (2016)*, and therefore, it is taken here as upper-bound threshold in performance.

## Simulations

A set of nonlinear toy examples typically used in clustering problems are proposed to test the different feature selection methods. In these experiments, the goal was to verify the correct selection of the original set of features. Figure 2 shows the toy examples considered[1], which are described by features $f_1$ and $f_2$, and the final description of the datasets can be seen in Table 2.

---

[1] The codes used to generate these datasets are available in https://github.com/avillago/u2fs.

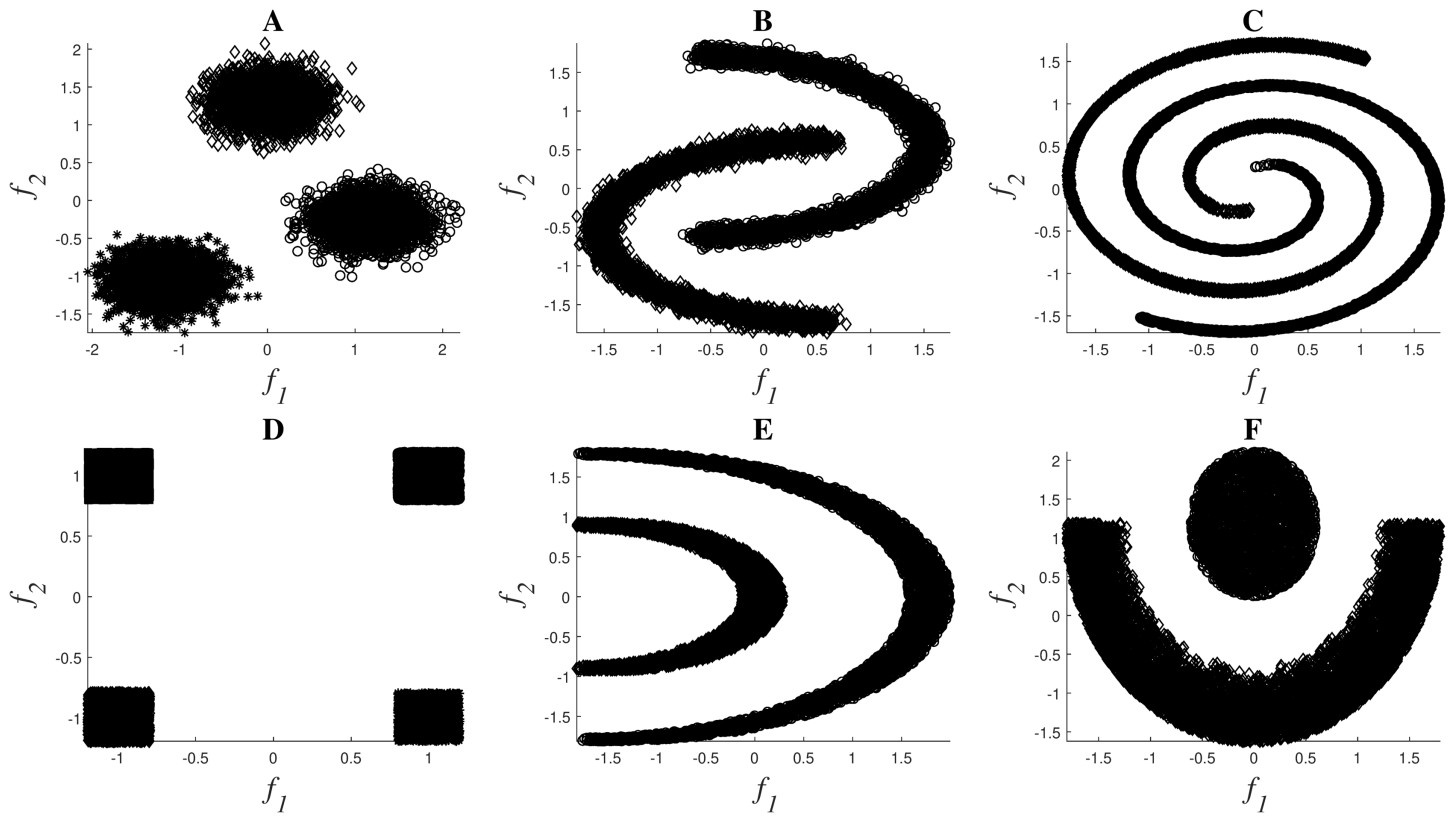

**Figure 2 Toy examples used for simulations: Clouds (A), Moons (B), Spirals (C), Corners (D), Half-Kernel (E), Crescent Moon (F).**

**Table 2 Description of the toy example datasets.**

|              | #samples | #classes |
| ------------ | -------- | -------- |
| Clouds       | 9,000    | 3        |
| Moons        | 10,000   | 2        |
| Spirals      | 10,000   | 2        |
| Corners      | 10,000   | 4        |
| Half-Kernel  | 10,000   | 2        |
| Crescent-Moon | 10,000  | 2        |

All these problems are balanced, except for the last dataset Cres-Moon, for which the data is divided 25% to 75% between the two clusters. Five extra features in addition to the original $f_1$ and $f_2$ were added to each of the datasets in order to include redundant or irrelevant information:

- $f_1'$ and $f_2'$: random values extracted from two Pearson distributions characterized by the same higher-order statistics as $f_1$ and $f_2$ respectively.
- $f_3'$ and $f_4'$: Original $f_1$ and $f_2$ contaminated with Gaussian noise ($v\mathcal{N}(0,1)$), with $v = 1.5$.
- $f_5'$: Constant feature of value 0.

**Table 3  Results feature selection for toy examples.** The results in bold correspond to the correct subset selection: $f1$ and $f2$.

| Method | Utility metric | | | $l_1 - norm$ | |
|---|---|---|---|---|---|
| Embedding | $KNN_{Bin}$ | $RBF_{\sigma_0^2}$ | $RBF_{\hat{\sigma}^2}$ | $KNN_{Bin}$ | $RBF_{\sigma_0^2}$ |
| Clouds | **$f1, f2$** | $f1, f4$ | **$f1, f2$** | $f1, f2$ | $f1, f2$ |
| Moons | **$f1, f2$** | $f3, f4$ | **$f1, f2$** | $f1, f3$ | $f1, f3$ |
| Spirals | **$f1, f2$** | **$f1, f2$** | **$f1, f2$** | $f2, f2$ | $f2, f2$ |
| Corners | **$f1, f2$** | $f1, f2$ | **$f1, f2$** | $f2, f2$ | $f2, f2$ |
| Half-Kernel | **$f1, f2$** | $f2, f3$ | **$f1, f2$** | $f1, f3$ | $f1, f3$ |
| Cres-Moon | **$f1, f2$** | $f1, f4$ | **$f1, f2$** | $f2, f1$ | $f2, f2$ |

The first step in the preprocessing of the features was to standardize the data using z-score to reduce the impact of differences in scaling and noise. In order to confirm the robustness of the feature selection techniques, the methods were applied using 10-fold cross-validation on the standardized data. For each fold a training set was selected using $m$-medoids, setting $m$ to 2,000 and using the centers of the clusters found as training samples. By doing so, the generalization ability of the methods can be guaranteed (*Varon, Alzate & Suykens, 2015*). On each of the 10 training sets, the features were selected applying the 5 methods mentioned in Table 1. For each of the methods, the number of clusters $c$ was introduced as the number of classes presented in Table 2. Since these experiments aim to evaluate the correct selection of the features, and the original features $f_1$ and $f_2$ are known, the number of features $s$ to be selected was set to 2.

Regarding the parameter settings within the embedding methods, the binary was obtained setting $k$ in the $k$NN approach to 5. For the RBF kernel embedding, $\sigma_0^2$ was set to the mean of the standard deviation along each dimension, as done in *Cai, Zhang & He (2010)*. When using $\hat{\sigma}^2$, its value was obtained by applying the method described in Algorithm 1.

In terms of subset selection approaches, the method based on the $l_1 - norm$ automatically sets the value of the regularization parameter required for the LARS implementation, as described in (*Cai & Chiyuan Zhang, 2020*). For the utility metric, $\beta$ was automatically set to the smallest non-zero eigenvalue of the matrix $\mathbf{R_{XX}}$ as described in Algorithm 2.

The performance of the algorithm is evaluated comparing the original set of features $f_1$ and $f_2$ to those selected by the algorithm. In these experiments, the evaluation of the selection results is binary: either the feature set selected is correct or not, regardless of the additional features $f'_i$, for $i = 1, 2, \ldots, 5$, selected.

In Table 3 the most common results obtained in the 10 folds are shown. The utility-based approaches always obtained the same results for all 10 folds of the experiments. On the contrary, the $l_1 - norm$ methods provided different results for different folds of the experiment. For these cases, Table 3 shows the most common feature pair for each experiment, occurring at least 3 times.

As shown in Table 3, the methods that always obtain the adequate set of features are based on utility, both with the binary weighting and with the RBF kernel and the suggested

**Table 4 Description of the benchmark databases.**

|  | Data type | Samples | Features | Classes |
|---|---|---|---|---|
| USPS | Images | 9,298 | 256 | 10 |
| Isolet | Audio | 1,560 | 617 | 26 |
| ORL | Images | 400 | 1,024 | 40 |
| COIL20 | Images | 1,440 | 1,024 | 20 |
| PCMAC | Text | 1,943 | 3,289 | 2 |
| BASEHOCK | Text | 1,993 | 4,862 | 2 |

$\hat{\sigma}^2$. Since these results were obtained for the 10 folds, they confirm both the robustness and the consistency of the U2FS algorithm.

## Benchmark datasets

Additionally, the proposed methods were evaluated using 6 well-known benchmark databases. The databases considered represent image (USPS, ORL, COIL20), audio (ISOLET) and text data (PCMAC, BASEHOCK)[2], proposing examples with more samples than features, and vice versa. The description of these databases is detailed in Table 4. All these datasets are balanced, except USPS.

In these datasets, the relevant features are unknown. Therefore, the common practice in the literature to evaluate feature selectors consists of applying the algorithms, taking from 10% to 80% of the original set of features, and evaluating the accuracy of a classifier when trained and evaluated with the selected feature set (*Zhu et al., 2016*). The classifier used for this aim in other papers is $k$-Nearest Neighbors (KNN), setting the number of neighbors to 5.

These accuracy results are computed using 10-fold cross-validation to confirm the generalization capabilities of the algorithm. By setting $m$ to 90% of the number of samples available in each benchmark dataset, $m$-medoids is used to select the $m$ centroids of the clusters and use them as training set. Feature selection and the training of the KNN classifier are performed in these 9 folds of the standardized data, and the accuracy of the KNN is evaluated in the remaining 10% for testing. Exclusively for USPS, given the size of the dataset, 2,000 samples were used for training and the remaining data was used for testing. These 2,000 samples were also selected using $m$-medoids. Since PCMAC and BASEHOCK consist of binary data, these datasets were not standardized.

The parameters required for the binary and RBF embeddings, as well as $\beta$ for the utility algorithm, are automatically set as detailed in "Discussion".

Figure 3 shows the median accuracy obtained for each of the 5 methods. The shadows along the lines correspond to the 25 and 75 percentile of the 10 folds. As a reference, the accuracy of the classifier without using feature selection is shown in black for each of the datasets. Additionally, Fig. 4 shows the computation time for both the utility metric and the $l_1 - norm$ applied on a binary weighting embedding. In this manner, the subset selection techniques can be evaluated regardless of the code efficiency of the embedding stage. Similarly to Fig. 3, the computation time plots show in bold the median running time

[2] All datasets downloaded from http://featureselection.asu.edu/datasets.php.

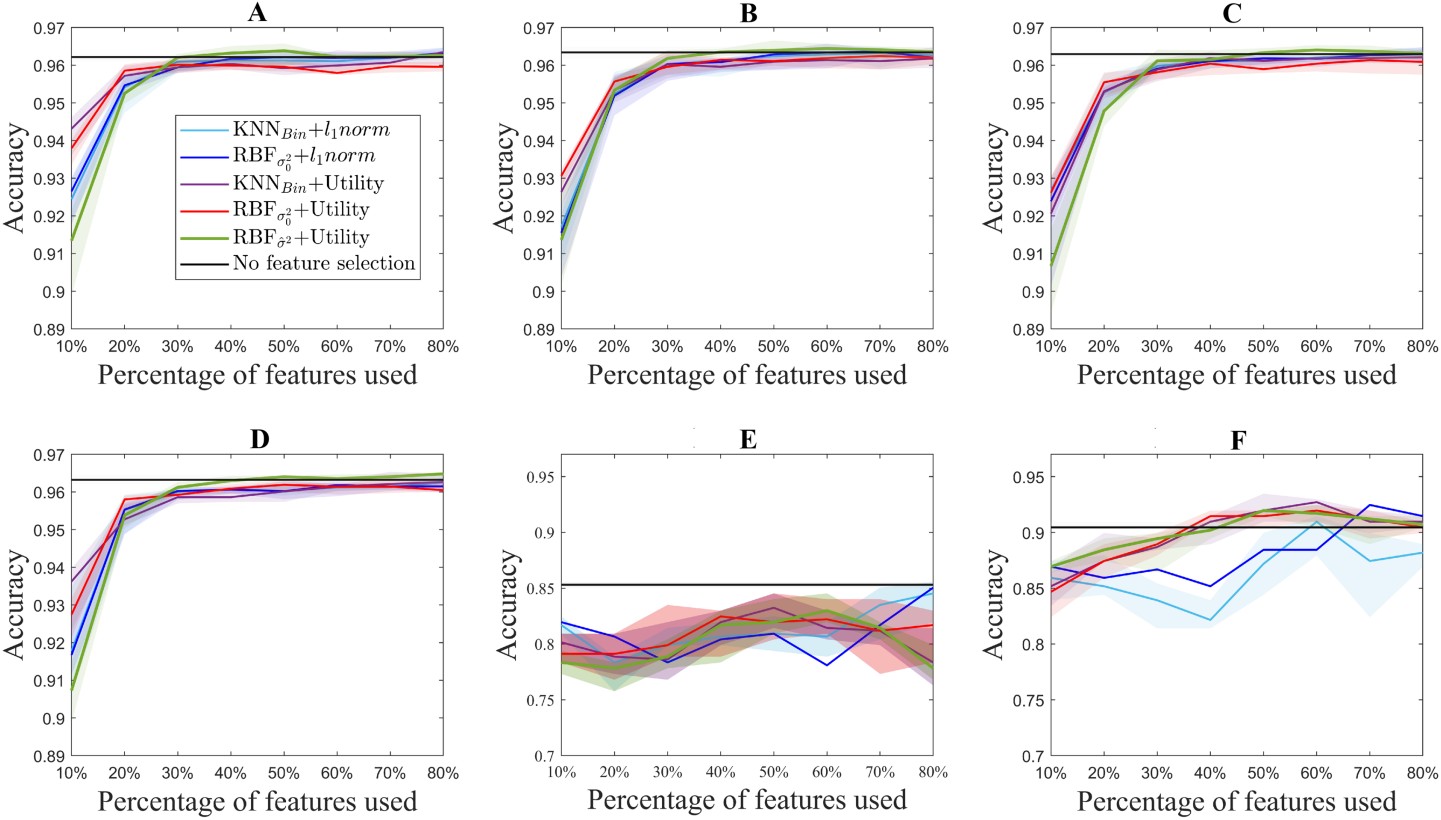

**Figure 3 Accuracy results for the benchmark databases, for selecting from 10% to 80% of the original number of features.** The thick lines represent the median accuracy of the 10-fold cross-validation, and the shadows, the 25 and 75 percentile. USPS (A), Isolet (B), ORL (C), COIL20 (D), PCMAC (E), BASEHOCK (F).

for each of the subset selection techniques, and the 25 and 75 percentiles around it obtained from the 10-fold cross-validation.

The difference in the trends of the $l_1 - norm$ and utility in terms of computation time is due to their formulation. Feature selection based on $l_1 - norm$ regularization, solved using the LARS algorithm in this case, requires the same computation time regardless of the number of features aimed to select. All features are evaluated together, and later on, an MCFS score obtained from the regression problem is assigned to them (*Cai, Zhang & He, 2010*). The features with the higher scores are the ones selected. On the other hand, since the utility metric is applied in a backward greedy trend, the computation times change for different number of features selected. The lower the number of features selected compared to the original set, the higher the computation time. This is aligned with the computational complexity of the algorithm, described in "Related Work". In spite of this, it can be seen that even the highest computation time for utility is lower than the time taken using $l_1 - norm$ regularization. The experiments were performed with 2x Intel Xeon E5-2640 @ 2.5 GHz processors and 64GB of working memory.

Finally, the experiments in benchmark databases are extended to compare U2FS to other key algorithms in the state-of-the-art. As it was mentioned at the beginning of this section, the selected algorithms are MCFS, NDFS, and RJGSC, which represent,

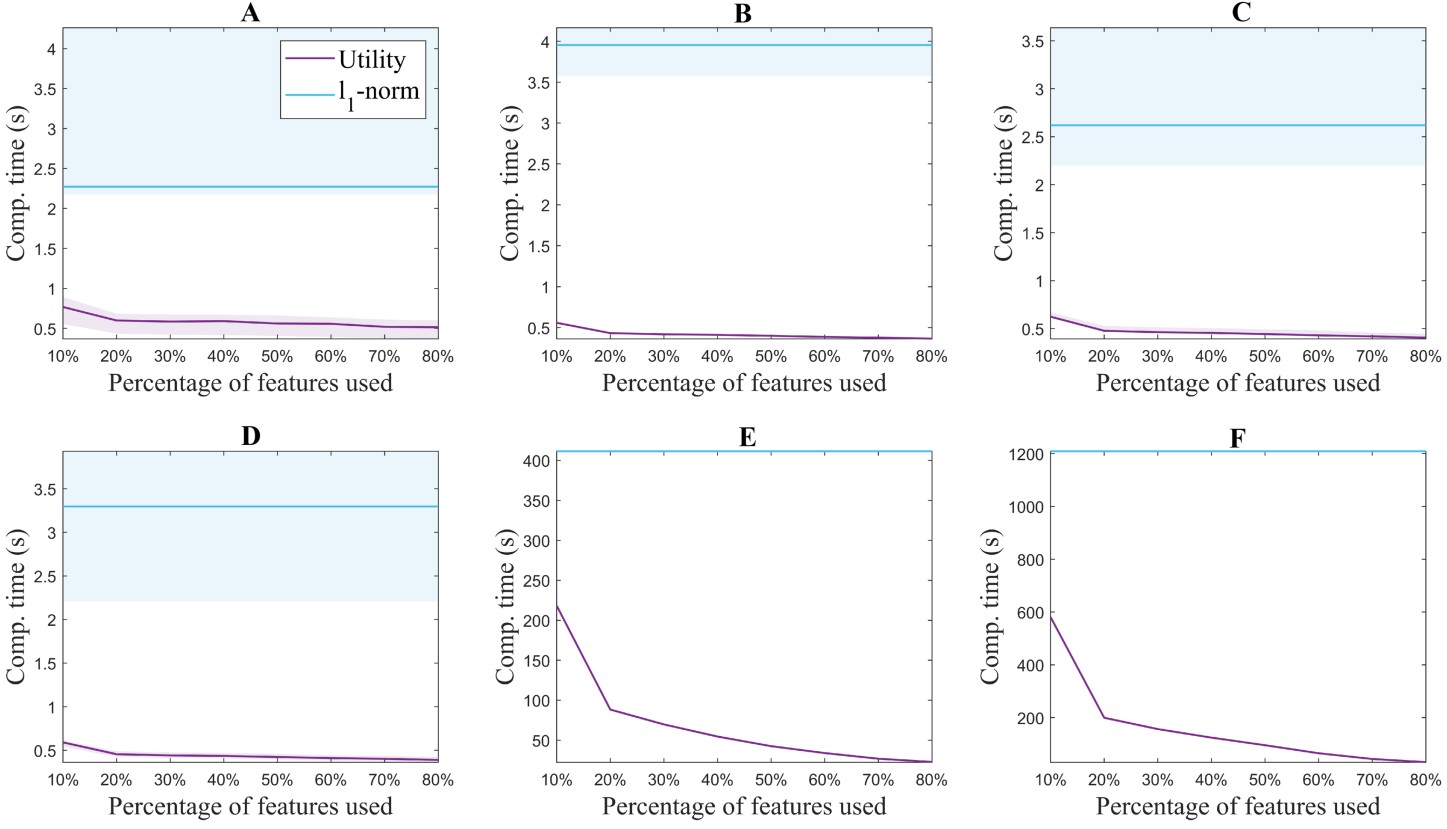

**Figure 4 Computation time for extracting from 10% to 80% of the original number of features for each of the benchmark databases. USPS (A), Isolet (B), ORL (C), COIL20 (D), PCMAC (E), BASEHOCK (F).**

respectively, the precursor of U2FS, an improved version of MCFS, and an example from the class of adaptive algorithms which recursively optimize the objective function proposed. NDFS and RJGSC require the tuning of their regularization parameters, for which the indications in their corresponding articles were followed. For NDFS, the value of $\gamma$ was set to $10^8$, and $\alpha$ and $\beta$ were selected from the values $\{10^{-6}, 10^{-4}, \ldots, 10^6\}$ applying grid search. The matrix F was initialized with the results of spectral clustering using all the features. For RJGSC, the results described in *Zhu et al. (2016)* for the BASEHOCK and PCMAC datasets are taken as a reference. In MCFS, the embedding is done using KNN and binary weighting, and the $l_1 - norm$ is used for subset selection. U2FS, on the other hand, results from the combination of the RBF kernel with $\hat{\sigma}^2$ and the utility metric. Table 5 summarizes the results by showing the KNN accuracy (ACC) for 10% of the features used, and the maximum ACC achieved among the percentages of features considered, for the BASEHOCK and PCMAC datasets.

## DISCUSSION

The results obtained in the experiments suggest that the proposed U2FS algorithm obtains comparable results to the state-of-the-art in all the applications suggested, taking less computational time. Nevertheless, the performance of the utility metric for feature selection varies for the different experiments presented and requires a detailed analysis.

**Table 5 Comparison of classification accuracy (ACC) with the state-of-the-art for PCMAC and BASEHOCK datasets.**

| Dataset | Method | ACC at 10% features | % features at Max ACC | Max ACC [b] |
|---------|--------|---------------------|-----------------------|-------------|
| PCMAC | U2FS | 0.785 | 60% | 0.83 |
| | MCFS | 0.67 | 20% | 0.697 |
| | NDFS | 0.73 | 40% | 0.83 |
| | RJGSC | 0.805 | 60% | 0.83 |
| BASEHOCK | U2FS | 0.87 | 50% | 0.925 |
| | MCFS | 0.815 | 80% | 0.84 |
| | NDFS | 0.76 | 20% | 0.794 |
| | RJGSC | 0.902 | 80% | 0.917 |

From Table 3, in "Discussion", it can be concluded that the utility metric is able to select the correct features in an artificially contaminated dataset. Both the binary embedding and the RBF kernel with $\hat{\sigma}^2$ select the original set of features for the 10 folds of the experiment. The stability in the results also applies for the RBF embedding with $\sigma_0^2$, which always selected the same feature pair for all 10 folds even though they are only correct for the spirals problem.

Therefore, considering the stability of the results, it can be concluded that the proposed approach is more robust in the selection of results than that based on the $l_1 - norm$.

On the other hand, when considering the suitability of the features selected, two observations can be made. First of all, it can be seen that the lack of consistency in the $l_1 - norm$ approaches discards the selection of the correct set of features. Moreover, the wrong results obtained with both $l_1 - norm$ and utility methods for the RBF embedding using $\sigma_0^2$ reveal the drawback of applying this approximation of $\sigma_0^2$ in presence of redundant or irrelevant features. Since this value is calculated as the mean of the standard deviation of all the dimensions in the data, this measure can be strongly affected by irrelevant data, that could be very noisy and enlarge this sigma, leading to the allocation of all the samples to a mega-cluster.

While the use of the proposed approximation for $\hat{\sigma}^2$ achieves better results than $\sigma_0^2$, these are comparable to the ones obtained with the KNN binary embedding when using the utility metric. The use of KNN to build graphs is a well-known practice, very robust for dense clusters, as it is the case in these examples. The definition of a specific field where each of the embeddings would be superior is beyond the scope of this paper. However, the excellence of both methods when combined with the proposed subset selection method only confirms the robustness of the utility metric, irrespective of the embedding considered.

For standardization purposes, the performance of the method was evaluated in benchmark databases. As it can be observed, in terms of the accuracy obtained for each experiment, U2FS achieves comparable results to the $l_1 - norm$ methods for most of the datasets considered, despite its condition of greedy method.

In spite of this, some differences in performance can be observed in the different datasets. The different ranking of the methods, as well as the accuracy obtained for each of the databases can be explained taking into account the type of data under study and the ratio between samples and dimensions.

With regard to the type of data represented by each test, it can be observed that for the ISOLET dataset, containing sound information, two groups of results are distinguishable. The group of the utility metric results outperforms those derived from the $l_1 - norm$, which only reach comparable results for 60% of the features selected. These two groups of results are caused by the subset selection method applied, and not for the embedding, among which the differences are not remarkable.

In a similar way, for the case of the image datasets USPS, ORL and COIL20, the results derived from utility are slightly better than those coming from the $l_1 - norm$. In these datasets, similarly to the performance observed in ISOLET, accuracy increases with the number of features selected.

Regarding the differences between the proposed embeddings, it can be observed that the results obtained are comparable for all of them. Nonetheless, Fig. 3 shows that there is a slight improvement in the aforementioned datasets for the RBF kernel with $\hat{\sigma}^2$, but the results are still comparable to those obtained with other embeddings. Moreover, this similarity in the binary and RBF results holds for the $l_1 - norm$ methods, for which the accuracy results almost overlap in Fig. 3. This can be explained by the relation between the features considered. Since for these datasets the samples correspond to pixels, and the features to the color codes, a simple neighboring method such as the binary weighting is able to code the connectivity of pixels of similar colors.

The text datasets, PCMAC and BASEHOCK, are the ones that show bigger differences between the results obtained with utility and those obtained with the $l_1 - norm$. This can be explained by the amount of zeros present in the data, with which the utility metric is able to cope slightly better. The sparsity of the data leads to more error in the $l_1 - norm$ results, since more features end up having the same MCFS score, and among those, the order for selection comes at random. The results obtained with the utility metric are more stable, in particular for the BASEHOCK dataset. For this dataset, U2FS even outperforms the results without feature selection if at least 40% of the features are kept.

In all the datasets proposed, the results obtained with the $l_1 - norm$ show greater variability, i.e., larger percentiles. This is aligned with the results obtained in the simulations. The results for the $l_1 - norm$ are not necessarily reproducible in different runs, since the algorithm is too sensitive to the training set selected. The variability of the utility methods is greater for the approaches based on the RBF kernel. This is due to the selection of the $\sigma^2$ parameter, which also depends on the training set. The tuning of this parameter is still very sensitive to high-dimensional and large-scale data, posing a continuous challenge for the machine learning community (*Yin & Yin, 2016*; *Tharwat, Hassanien & Elnaghi, 2017*).

Despite it being a greedy method, the utility metric proves to be applicable to feature selection approaches and to strongly outperform the $l_1 - norm$ in terms of computational

time, without significant reduction in accuracy. U2FS proves to be effective both in cases with more samples than features and vice versa. The reduction in computation time is clear, for all the benchmark databases described, and is particularly attractive for high-dimensional datasets. Altogether, our feature selection approach U2FS, based on the utility metric, and with the binary or the RBF kernel with $\hat{\sigma}^2$ is recommended due to its fast performance and its interpretability.

Additionally, the performance of U2FS is comparable to the state-of-the-art, as shown in Table 5. In this table, the performance of U2FS (RBF kernel and $\hat{\sigma}^2$, with the utility metric) is compared to that of MCFS, NDFS and RJGSC. For MCFS, it can be seen that, as expected, U2FS appears as an improvement of this algorithm, achieving better results for both datasets. For NDFS, the results are slightly worse than for U2FS, most probably due to problems in the tuning of regularization parameters. Given the consistent good results for different datasets of RJGSC when compared against the state-of-the-art, and its condition of simultaneously adapting the spectral embedding and subset selection stages, this algorithm is taken as example of the most complex SSFS algorithms (SAMM-FS, SOGFS or DSRMR). These algorithms perform manifold learning and feature selection simultaneously, iteratively adapting both steps to achieve optimal results.

It is clear that in terms of accuracy, both for 10% of the features and for the maximal value of achieved, U2FS obtains similar results to RJGSC, while at the same time having a much smaller computational complexity. Furthermore, while RJGSC requires the manual tuning of extra parameters, similarly to other algorithms in the state-of-the-art, U2FS tunes its parameters automatically. Hence, the application of the method is straightforward for the users. The stages of higher complexity in U2FS, previously defined as $O(N^3 + d^3)$, are shared by most of the algorithms in the state-of-the-art. However, on top of these eigendecompositions and matrix inversions, the algorithms in the literature require a number of iterations in the optimization process that U2FS avoids. Additionally, U2FS is the only algorithm for which the computation time scales linearly with the amount of features selected.

The current state-of-the-art of unsupervised spectral feature selectors applies the stages of manifold learning and subset selection simultaneously, which can lead to optimal results. In a field that gets more and more complex and goes far from applicability, U2FS is presented as a quick solution for a sequential implementation of both stages of SSFS algorithms, yet achieving comparable results to the state-of-the-art. Being a greedy method, the utility metric cannot be applied simultaneously to the manifold learning and subset selection stages. However, other sequential algorithms from the state-of-the-art could consider the use of utility for subset selection, instead of the current sparsity-inducing techniques. One of the most direct applications could be the substitution of group-LASSO for group-utility, in order to perform selections of groups of features as proposed by *Bertrand (2018)*. This can be of interest in cases where the relations between features are known, such as in channel selection (*Narayanan & Bertrand, 2020*) or in multi-modal applications (*Zhao, Hu & Wang, 2015*).

## CONCLUSION

This work presents a new method for unsupervised feature selection based on manifold learning and sparse regression. The main contribution of this paper is the formulation of the utility metric in the field of spectral feature selection, substituting other sparse regression methods that require more computational resources. This method, being a backward greedy approach, has been proven to obtain comparable results to the state-of-the-art methods with analogous embedding approaches, yet at considerably reduced computational load. The method shows consistently good results in different applications, from images to text and sound data; and it is broadly applicable to problems of any size: using more features than samples or vice versa.

Furthermore, aiming to show the applicability of U2FS to data presenting non-linearities, the proposed approach has been evaluated in simulated data, considering both a binary and an RBF kernel embedding. Given the sensitivity of the RBF kernel to high-dimensional spaces, a new approximation of the RBF kernel parameter was proposed, which does not require further tuning around the value obtained. The proposed approximation outperforms the rule-of-thumb widely used in the literature in most of the scenarios presented. Nevertheless, in terms of feature selection, the utility metric is robust against the embedding.

U2FS is proposed as a non-parametric efficient algorithm, which does not require any manual tuning or special knowledge from the user. Its simplicity, robustness and accuracy open a new path for structure sparsity-inducing feature selection methods, which can benefit from this quick and efficient technique.

### Funding

This work received funding from FWO project G0A4918N. This project received funding from the European Research Council (ERC) under the European Union's Horizon 2020 research and innovation programme (grant agreement No. 802895). This research received funding from the Flemish Government (AI Research Program). This work was supported by Bijzonder Onderzoeksfonds KU Leuven (BOF): The effect of perinatal stress on the later outcome in preterm babies: C24/15/036, Prevalentie van epilepsie en slaapstoornissen in de ziekte van Alzheimer: C24/18/097. Agentschap Innoveren en Ondernemen (VLAIO) 150466: OSA+ and O\& O HBC 2016 0184 eWatch. KU Leuven Stadius acknowledges the financial support of imec, and EU H2020 MSCA-ITN-2018: INtegrating Magnetic Resonance SPectroscopy and Multimodal Imaging for Research and Education in MEDicine (INSPiRE-MED), funded by the European Commission under Grant Agreement no. 813120. EU H2020 MSCA-ITN-2018: 'INtegrating Functional Assessment measures for Neonatal Safeguard (INFANS)', funded by the European Commission under Grant Agreement no. 813483. EIT 19263–SeizeIT2: Discreet Personalized Epileptic Seizure Detection Device. The resources and services used in the experiments of this work were provided by the VSC (Flemish Supercomputer Center), funded by the Research Foundation—Flanders (FWO) and the Flemish Government. There was no additional

external funding received for this study. The funders had no role in study design, data collection and analysis, decision to publish, or preparation of the manuscript.

## Grant Disclosures

The following grant information was disclosed by the authors:

FWO project: G0A4918N.

European Research Council (ERC): 802895.

Flemish Government: AI Research Program.

Bijzonder Onderzoeksfonds KU Leuven (BOF): C24/15/036 and C24/18/097.

Agentschap Innoveren en Ondernemen (VLAIO): 150466.

European Commission: 813120 and 813483.

## Competing Interests

The authors declare that they have no competing interests.

## Author Contributions

- Amalia Villa conceived and designed the experiments, performed the experiments, analyzed the data, performed the computation work, prepared figures and/or tables, authored or reviewed drafts of the paper, and approved the final draft.
- Abhijith Mundanad Narayanan performed the computation work, authored or reviewed drafts of the paper, and approved the final draft.
- Sabine Van Huffel conceived and designed the experiments, authored or reviewed drafts of the paper, and approved the final draft.
- Alexander Bertrand conceived and designed the experiments, authored or reviewed drafts of the paper, and approved the final draft.
- Carolina Varon conceived and designed the experiments, analyzed the data, performed the computation work, authored or reviewed drafts of the paper, and approved the final draft.

## Data Availability

We provide the code of the U2FS algorithm, which includes a code for approximating the RBF kernel parameter, and the utility metric for subset selection. Additionally, a code to generate our simulated data is provided, as well as an example script on how to use the codes.

The codes can be accessed at GitHub: https://github.com/avillago/u2fs.

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
