# Peer review of "Utility metric for unsupervised feature selection"

_PeerJ Computer Science, doi:10.7717/peerj-cs.477_

## Round 0.1 · original submission · Major Revisions

Reviewers have commented on your research paper. They found the study to be of interest to the research community, but with the problem of an unfair experimental setup in terms of comparison wrt the state-of-the-art feature extraction methods. There are also several minor remarks that must be addressed prior to publication.

Reviewer 1 ·

Basic reporting

The writing of the paper requires significant revisions, particularly in technical aspects.

Experimental design

The experiments lack the comparison with state-of-the-art unsupervised feature selection (dimensional reduction) methods, i.e., Graph-based methods, Extended Sammon projection and wavelet kernel extreme learning machine for gait-based legitimate user identification, Proceedings of the 34th ACM/SIGAPP Symposium on Applied Computing, 1216-1219. Etc.

Validity of the findings

It's a good point that the authors have provided the code for readers to prove the claims, which is quite helpful for the community to use the technique for future studies.

Additional comments

The writing of the paper requires significant revisions, particularly in technical aspects. The experiments lack the comparison with state-of-the-art unsupervised feature selection (dimensional reduction) methods, i.e., Graph-based methods, Extended Sammon projection and wavelet kernel extreme learning machine for gait-based legitimate user identification, Proceedings of the 34th ACM/SIGAPP Symposium on Applied Computing, 1216-1219. Etc. It's a good point that the authors have provided the code for readers to prove the claims, which is quite helpful for the community to use the technique for future studies.

Reviewer 2 ·

Basic reporting

In this manuscript the authors presented U2FS a novel method for automated feature selection which is important step of dimensional reduction in machine learning. The U2FS is an unsupervised feature selection method which belongs to spectral/sparse learning category. The authors designed new utility metric which is the basis for the U2FS algorithm and is computational efficient. They also explored the use of radial basic function (RBF) kernel and k-nearest neighbours (KNN) algorithms in the first structure extraction learning stage of feature selection.
The main advantages of proposed method are, comparing to other state-of-the art methods, it does not require tuning of the parameters and has less computational time, while its performance is comparable to them in terms of accuracy. Also authors published their method, implemented in Matlab code, as a free publicly available source via github repository.
The article is written in clear and correct English language. The sections Introduction, Related work and Methods provide sufficient and detailed information and relevant literature about the article’s topic. Manuscript has standard structure, and figures and tables are in appropriate format and resolution. Formulas and algorithms of the introduced metrics and methods are provided in details. All results are relevant to hypothesis, with major suggestion that proposed algorithm should be compared to more state-of-the-art methods and some additional evaluation measures could be used.

Experimental design

Research question is well defined and relevant. The methods are described in details with sufficient information to be reproducible. There is a major suggestion about the investigation, that the novel method should be tested with additional evaluation measures and compared to a wider set of state-of-the-art methods. Also authors should emphasize that the analysis included only methods within the same category as the investigated novel U2FS method.

Validity of the findings

The URLs of the datasets used in the research are provided correctly and code for generation of simulation sets and implementation of the method is given on github repository. The minor suggestion is that Matlab code for generation of simulation samples for two sets of Moons and Clouds is missing from the github repository. The experiments are well and in details preformed and conclusions are well stated. However regarding the conclusion about the comparability of novel method with state-of-the-arts in terms of accuracy in the section Results (page 12 and Table 5), my major suggestions are to use additional evaluation measure e.g. error rate, beside the classification accuracy, and to include in the benchmark part more state-of-the-art methods (of the same type and subcategory as U2FS, e.g. NDFS [Li Z, Yang Y, Liu J, Zhou X, Lu H. Unsupervised feature selection using nonnegative spectral analysis. In Twenty-Sixth AAAI Conference on Artificial Intelligence 2012 Jul 14.], DSRMR [Tang C, Liu X, Li M, Wang P, Chen J, Wang L, Li W. Robust unsupervised feature selection via dual self-representation and manifold regularization. Knowledge-Based Systems. 2018 Apr 1;145:109-20.]) beside the RJGSC and MCFS methods to compare U2FS with.

Additional comments

The major suggestion:
1. The evaluation of the proposed U2FS method by comparing to other state-of-the-art methods in section Results (page 12 and Table 5) should be expended to more methods (of the same type and subcategory as U2FS, e.g. NDFS [Li Z, Yang Y, Liu J, Zhou X, Lu H. Unsupervised feature selection using nonnegative spectral analysis. In Twenty-Sixth AAAI Conference on Artificial Intelligence 2012 Jul 14.], DSRMR [Tang C, Liu X, Li M, Wang P, Chen J, Wang L, Li W. Robust unsupervised feature selection via dual self-representation and manifold regularization. Knowledge-Based Systems. 2018 Apr 1;145:109-20.]) beside the RJGSC and MCFS methods to compare U2FS with.

The minor suggestions:
1. In the section Results in the evaluation analysis, beside the classification accuracy, additional evaluation measure e.g. error rate could be used.
2. The URL of the github method could be more explicit given within the manuscript text itself, because it is the valuable result of the work.
3. The README file on the github could be more informative, e.g. basic description of the U2FS method, requirement list if any, example of basic use of the code, etc.
4. The data i.e. Matlab code for generation for simulation samples for two sets of Moons and Clouds is missing from the github repository.
5. In the section Results, in the ‘comparison with state-of-the-art methods’ part (page 12), it should be emphasized in the manuscript that analysis included only methods within the same category as investigated novel U2FS method.

---

## Round 0.2 · Minor Revisions

The current version of the paper is fine, but authors must address comments provided by reviewer #3

Reviewer 1 ·

Basic reporting

N.A

Experimental design

N.A

Validity of the findings

N.A

Additional comments

Thank you for addressing the comments. I have no further comments. Good luck!

Reviewer 3 ·

Basic reporting

NO COMMENT

Experimental design

NO COMMENT

Validity of the findings

NO COMMENT

Additional comments

1.The sample code in the manuscript code will encounter dimension errors. Please check the experiment code again.
2.The comparison method and the data set used can be increased, which will make the article more professional and persuasive.
3.The logic in the manuscript is clear, the grammar is accurate, and the format is relatively standardized.
The formula is best placed in the center to look more tidy.

---

## Round 0.3 · accepted · Accept

Authors have successfully addressed all comments provided by reviewers and therefore manuscript is ready for publication.